# *Arthrocnemum* Moq.: Unlocking Opportunities for Biosaline Agriculture and Improved Human Nutrition

**DOI:** 10.3390/plants13040496

**Published:** 2024-02-09

**Authors:** Esteban Ramírez, Nuria Rodríguez, Vicenta de la Fuente

**Affiliations:** 1Departamento de Biología, Facultad de Ciencias, Universidad Autónoma de Madrid, Cantoblanco, 28049 Madrid, Spain; 2Centro de Astrobiología (CAB), CSIC-INTA, Torrejón de Ardoz, 28850 Madrid, Spain; rodriguezgn@cab.inta-csic.es

**Keywords:** *Arthrocnemum*, elemental composition, biominerals, succulent stems, seeds

## Abstract

(1) Background: This study provides novel insights into the elemental content and biomineralization processes of two halophytic species of the genus *Arthrocnemum* Moq. (*A. macrostachyum* and *A. meridionale*). (2) Methods: Elemental content was analyzed using ICP-MS, while biominerals were detected through electron microscopy (SEM and TEM) and X-ray diffraction. (3) Results: The elemental content showed significant concentrations of macronutrients (sodium, potassium, magnesium, and calcium) and micronutrients, especially iron. Iron was consistently found as ferritin in *A. macrostachyum* chloroplasts. Notably, *A. macrostachyum* populations from the Center of the Iberian Peninsula exhibited exceptionally high magnesium content, with values that exceeded 40,000 mg/kg d.w. Succulent stems showed elemental content consistent with the minerals identified through X-ray diffraction analysis (halite, sylvite, natroxalate, and glushinskite). Seed analysis revealed elevated levels of macro- and micronutrients and the absence of heavy metals. Additionally, the presence of reduced sodium chloride crystals in the seed edges suggested a mechanism to mitigate potential sodium toxicity. (4) Conclusions: These findings highlight the potential of *Arthrocnemum* species as emerging edible halophytes with nutritional properties, particularly in Western European Mediterranean territories and North Africa. They offer promising prospects for biosaline agriculture and biotechnology applications.

## 1. Introduction

Halophytes, which are plants that thrive and reproduce in environments with high saline concentrations (≥200 mM NaCl), are commonly found in both coastal and inland salt marshes. They contribute to approximately 1% of the world’s flora diversity [1]. However, the eHALOPH database, a well-known resource, includes records of plant species capable of tolerating lower saline concentrations starting at 80 mM NaCl [2]. This expanded range of tolerance increases the number of species that can successfully inhabit saline ecosystems.

The microhabitat of each halophyte is influenced by various ecological factors. These factors include the concentration of salts in the water and soil, as well as soil properties such as electrical conductivity, pH, moisture, and texture. Additionally, the daily and seasonal variations caused by tidal influences in coastal environments and/or the degree of flooding in inland salt marshes also impact the microhabitat. These environmental factors ultimately determine the presence and dominance of different taxa across continents. Furthermore, high temperatures and seasonal droughts are significant factors in arid and semiarid ecosystems [3,4], as well as in regions influenced by a Mediterranean macrobioclimate. These climatic conditions further shape the distribution and survival of halophytes in these areas.

Throughout their evolutionary history, halophytes have developed distinct anatomical, physiological, and metabolic adaptations to cope with excess salts. These adaptations include exclusion mechanisms, glandular secretion, succulence, and the formation of biominerals within their cells and tissues [5,6,7,8,9]. Maintaining a positive turgor pressure is crucial for halophytes, necessitating proper osmotic adjustment. To achieve this, these plants accumulate saline ions (primarily Na^+^ and Cl^−^ in eudicotyledons) and organic solutes within their vacuoles and cell cytoplasm. These organic solutes, known as osmolytes, encompass a range of compounds such as amino acids (e.g., proline, glycine, betaine), soluble sugars (e.g., glucose, fructose, sucrose), and polyalcohols (e.g., sorbitol, mannitol, glycerol, inositol), among others [10].

The cultivation of halophilic plants holds great potential for advancing biosaline agriculture, as highlighted by Grigore and Vicente [11]. The increasing prevalence of stress conditions such as drought, salinity, and high temperatures has led to a growing interest in utilizing wild halophytes as crops. These crops, known for their tolerance to salinity, are proving to be a valuable alternative for human nutrition and various biotechnological applications [12,13,14].

Within the Amaranthaceae/Chenopodiaceae family, the Salicornioideae subfamily stands out as a group of halophytes that have successfully adapted to coastal and inland saline habitats. This subfamily comprises 12 genera and approximately 100 species [15,16]. Among these, the annual species of *Salicornia* L. have gained significant attention as a widely utilized edible food source [17,18,19,20,21,22]. However, ongoing research is exploring the potential of other perennial halophytes such as *Sarcocornia* A.J. Scott, and *Arthrocnemum* Moq. as emerging crops. These plant species exhibit higher biomass production and demonstrate the ability to thrive in soils with high salt concentrations [23,24].

The Mediterranean region is home to two species of the genus *Arthrocnemum*, which include the Spanish and Portuguese Atlantic coasts, the eastern Mediterranean European territories, and the North African territories. These species are *Arthrocnemum macrostachyum* (Moric.) K. Koch, which has been recognized in Northern Italy, France, Spain, and Portugal, and *Arthrocnemum meridionale* (Ramírez et al.) Fuente et al., which has recently been recorded in the Italian islands of Sardinia and Sicily, Malta, Tunisia, Turkey, and Iran [25]. Our research team, focused on halophytes, is currently investigating the global distribution and biogeographical limits of both taxa.

Both *A. macrostachyum* and *A. meridionale* are considered emerging agricultural halophytes with promising biotechnological applications. Numerous authors have highlighted various phytochemical properties found in their extracts and tissues, which have potential nutritional and health benefits for humans [26,27,28,29,30]. Furthermore, these species could play a crucial role in saline agriculture by providing feed for livestock such as camels, sheep, or horses, among others [31,32,33]. Halophytes are of great interest to the agricultural industry because most non-halophilic vascular plants exclude sodium from their shoots and leaves, making it one of the most limiting minerals in the diet of herbivores [34].

The scientific literature suggests that *Arthrocnemum* plants hold great promise for various biotechnological applications due to their rich diversity of bioactive compounds, particularly phenolic compounds and fatty acids. These compounds have demonstrated notable antioxidant capacity [35]. Moreover, *Arthrocnemum* has been identified as a particularly interesting halotolerant species for bioenergy production, owing to the high quantity and quality of oil present in its seeds [36].

Another intriguing characteristic of this plant genus is its capacity to accumulate inorganic and/or mineral salts within its tissues [26,37]. Specifically, *A. macrostachyum* samples collected from Spain’s Tinto River in Huelva and Portugal’s Algarve region showcased a remarkable nutrient content in their succulent stems, primarily composed of sodium (Na), potassium (K), calcium (Ca), magnesium (Mg), iron (Fe), manganese (Mn), and zinc (Zn). Furthermore, research has indicated that the accumulation of ions can lead to the formation of biominerals in all plant tissues [38].

Among the most frequently observed biominerals in plants are magnesium, silicon, and iron oxalates, in the forms of jarosite and Fe oxides [8,39,40]. In fact, de la Fuente et al. [6] identified halite and sylvite chlorides, as well as glushinskite and weddellite oxalates, in the succulent stem tissues of the coastal halophyte *Sarcocornia pruinosa* Fuente, Rufo, and Sánchez-Mata. Due to their substantial accumulation of salts and natural crystals within their succulent structures, species from the Salicornioideae subfamily have historically been employed in traditional soap manufacturing and have also been utilized in various combustion processes to produce high-quality glass [41]. However, the widespread use of these plants declined over time, and it is only recently that they have regained recognition as valuable biological resources with multiple potential applications.

This study aims to investigate the nutritional and biotechnological potential of *Arthrocnemum* species found in Mediterranean and Atlantic European populations, as well as North African populations. This research presents data on the elemental composition of the succulent stems and seeds of these plants, as well as the processes underlying their biomineralization.

## 2. Results

### 2.1. Vegetative and Floriferous Succulent Stems

The order of macronutrient accumulation in *A. macrostachyum* and *A. meridionale* is Na > K > Mg > Ca, while iron follows calcium in the accumulation gradient for micronutrients. Table 1 and Appendix A present the semiquantitative content for *A. macrostachyum*, while Table 2 and Appendix A represent the content for *A. meridionale*. Table 3 displays the ICP-MS values for seeds.

Significant statistical differences have been observed in the comparison of various elements (Na, K, Mg, Ca, Zn, B, Mn, Mo, Ni, Ba, and Sr) between the populations of the Tinto River (Southwestern Iberian Peninsula, SW), the Center of the Iberian Peninsula (C), and the East (E) for *A. macrostachyum* (Figure 1 and Figure 2). However, due to the geographical idiosyncrasy of each population analyzed, *A. meridionale* cannot be treated as a comparable homogeneous group. The data obtained for this species exhibit great variability, primarily influenced by the geographical locations of the populations analyzed (ST18: continental salt marsh from southern Tunisia; ST19: coastal salt flats on the island of Malta; ST20: coastal salt flats in Tunisia, far from the influence of the sea).

The sodium element exhibits high levels ranging from 57,747.82 mg/kg to 311,083.18 mg/kg. Comparing the populations from the center of Spain to those from the Tinto River and the East of the Iberian Peninsula (Figure 1), it is evident that the sodium content is significantly higher in the former. The potassium content shows notable variations depending on the geographic locations of the samples. Statistical analysis reveals no significant differences between the populations of the Tinto River and those of eastern Spain, with potassium levels ranging from 6150.54 mg/kg to 33,717.76 mg/kg. However, samples collected in the province of Toledo (Villasequilla de Yepes and Lillo) exhibit significantly higher potassium values, measuring 45,374.50 mg/kg and 64,870.74 mg/kg, respectively (Figure 1).

The sodium content in *A. meridionale* exhibited a range of values ranging from 49,275.46 mg/kg to 218,830.71 mg/kg, which closely resembled the values analyzed for *A. macrostachyum*. Similarly, potassium values showed similarity between both species, with variable values ranging from 4972.65 mg/kg to 36,941.86 mg/kg in the three samples of *A. meridionale* that were analyzed.

To examine the structure of the succulent photosynthetic stems in both species, scanning electron microscopy (SEM) was employed to observe the cross sections. The observations revealed a distinct arrangement of organs and tissues, progressing from the outermost layers to the innermost ones. This arrangement can be described as follows: cuticle, epidermis, palisade parenchyma, and water storage tissue. The water-storage parenchyma consists of large, sinuous cells with thin cell walls, constituting the majority of the stem’s volume. Occasional vascular elements are scattered throughout this parenchyma. Following this section is the cortical parenchyma, followed by the central cylinder and the pith. The central cylinder contains vascular tissues (phloem and xylem) as well as parenchyma cells (Figure 3).

The biomineralization study reveals that sodium is the predominant element present in both succulent vegetative stems and inflorescences. This is evident in the tissues and cells, as depicted in Figure 4 and Figure 5. Figure 4 highlights the abundant manifestation of sodium in various regions, including the epidermis, water storage tissue, cortical parenchyma, vascular bundles (xylem and phloem), and pith. Notably, the central cylinder occasionally experiences collapse due to amorphous accumulations of sodium (Na) and chlorine (Cl). These amorphous Na and Cl deposits, along with crystalline halite, are also observed in the inflorescences.

Interestingly, sodium and chlorine crystallizations are detected in the cross-section of the inflorescence, particularly in the external layers of the seeds during their maturation process (Figure 4B,C). Additionally, cubic halite crystallizations are observed in Figure 4D and Figure 5A. These findings shed light on the intricate mineral composition and distribution within the studied plant structures.

The dispersive energy X-ray microanalysis, in conjunction with a scanning electron microscope, has provided spectra indicating the presence of halite as the predominant element. Additionally, crystallizations of sodium and chlorine have been observed in certain cases (Figure 5B). Despite high potassium concentrations determined by ICP-MS, the occurrence of sylvite (KCl) crystallizations is less frequent compared to halite (Figure 5C,D). Furthermore, fibrous crystallization of sodium, carbon, oxygen, and sulfur has been detected in the analyzed samples (Figure 5E,F).

In terms of calcium content, the values in *A. macrostachyum* exhibit a similar order of magnitude, ranging from 1829.82 to 8162.98 mg/kg. However, significant differences can still be observed between the higher concentrations found in the eastern peninsular populations compared to those in the center and the Tinto River populations (Figure 1). There are two exceptions where an additional order of magnitude is observed: sample 13 from the center with a calcium concentration of 11,817.14 mg/kg, and sample 16 from the East with a concentration of 26,030.18 mg/kg. Regarding *A. meridionale*, the calcium content varies significantly among different plant samples. Particularly, one of the Tunisian populations (Table 2, sample ST18) exhibits considerably higher calcium content at 25,407.54 mg/kg.

Magnesium exhibits remarkable homogeneity across the 11 samples collected from the Tinto River (ST1 to ST11), ranging from 4739.91 to 9042.43 mg/kg. Statistical analysis indicates that these values are significantly lower compared to other geographically distinct populations. Interestingly, the populations analyzed from the Eastern Peninsula display significantly higher levels of magnesium, ranging from 7110.62 to 21,258.66 mg/kg. Moreover, samples from the central region (samples ST12 and ST13) exhibit significantly elevated magnesium content, ranging from 41,491.40 to 41,993.52 mg/kg (Figure 1). In the case of *A. meridionale*, this element demonstrates similarly high concentrations, ranging between 13,095.79 and 21,597.93 mg/kg. These values surpass all those observed in the populations of the Tinto River and more closely align with the intervals obtained from individuals of the *A. macrostachyum* species found in the Eastern part of the Iberian Peninsula.

Upon microscopic examination, calcium appears to be uniformly distributed within cells, organelles, and various tissues (Figure 6 and Figure 7). Calcium, an essential biological component, is observed in cells of all tissues (Figure 6B,D). Occasionally, the presence of calcium and sulfur crystals is detected in parenchymatic tissues and the central cylinder (Figure 6A,C). This vital element often manifests as crystallized calcium oxalates in cells and tissues. Figure 7 illustrates the diverse forms of this mineral, including prismatic, rectangular, polyhedral, and cubic structures.

Magnesium emerges as one of the prominent elements within the investigated samples of both species, permeating every cell and tissue. Profound accumulations of magnesium are observed in the epidermis, water storage tissue, and cortical parenchyma. The representative images depicted in Figure 8 unveil two distinct morphological structures, namely cubes and druses (Figure 8A–C). These structures correspond to magnesium oxalates or hydroxides and have been identified as glushinskite (Mg(C_2_O_4_).2H_2_O) via X-ray diffraction analysis. Furthermore, the spectra presented in Figure 8B,D,F validate the presence of all the constituent elements of the glushinskite mineral.

About the micronutrients (Fe, Zn, B, Mn, Mo, Cu, and Ni), the concentration of iron exhibited significant variation, ranging between 77.78 mg/kg and 1004.92 mg/kg, except for sample ST9 from the Tinto River estuary, which displayed a remarkably higher concentration (2135.60 mg/kg d.w.). Nonetheless, the iron content remained relatively consistent among the 11 samples obtained from the Tinto River, with no statistically significant variances observed among the different geographical populations under investigation (Figure 2). Notably, *A. meridionale* generally exhibited comparable or slightly elevated iron levels compared to *A. macrostachyum*, particularly evident in the two Tunisian populations (543.45–1833 mg/kg). The Maltese population, represented by sample S20 in Table 2, demonstrated an iron concentration within the typical range for *A. macrostachyum* (247.93 mg/kg).

The application of dispersion energy analysis through scanning electron microscopy (SEM) for iron detection yielded limited outcomes, as only sporadic iron accumulations were identified within the tissues employing this technique. Iron, a vital micronutrient, has been successfully detected using the transmission electron microscopy (TEM) technique. The Tinto River samples of *A. macrostachyum* revealed the presence of iron in the form of ferritin (Figure 9). This protein was abundantly localized within the chloroplasts of cells and tissues present in succulent stems and inflorescences (Figure 9A). In particular, the ferritin protein was observed within the stroma of chloroplasts (Figure 9B–D). Interestingly, occasional detection of multiple ferritin nuclei was also noted (Figure 9C).

In terms of the presence of other micronutrients not detectable through SEM microscopy, the zinc content demonstrated variability within a similar range across samples, ranging from 16.22 mg/kg to 79.74 mg/kg, except for sample ST9, which exhibited a higher concentration of 107.03 mg/kg. Boron content ranged between 23.15 mg/kg and 76.47 mg/kg in most samples, but significantly higher levels were observed in samples from the central region of the Iberian Peninsula. However, for samples ST15, ST16, and ST17 from East Spain, no boron values were obtained.

The manganese content varied between 6.25 mg/kg (sample ST11) and 173.69 mg/kg (sample ST12). Molybdenum concentration was generally low, except for samples ST9 (Tinto River estuary: 11.58 mg/kg) and ST12 (Villasequilla de Yepes: 10.34 mg/kg). For zinc, boron, manganese, and molybdenum, statistically significant differences were observed between the highest concentrations in populations from the central region compared to the lower concentrations in populations analyzed from the Tinto River and eastern Spain (Figure 1). Copper ranged from 5.92 mg/kg to 72.38 mg/kg, whereas nickel exhibited a range from very low concentrations (0.37 mg/kg) to higher values (14.71 mg/kg), with significant differences observed for nickel between the higher concentrations in populations from the central and southwestern regions compared to the lower concentrations in the populations from the eastern region of the Iberian Peninsula (Figure 2).

The micronutrient composition of *A. meridionale*, encompassing elements such as Zn, B, Mn, Mo, Cu, and Ni, exhibited analogous values to those discerned in *A. macrostachyum* specimens. Notably, the ICP-MS analysis failed to recover values for boron in *A. meridionale* samples.

Concerning alkaline earth metals, the concentration of barium (ranging from 0.77 to 15.32 mg/kg) was observed to be comparatively lower than that of strontium (ranging from 13.62 to 146.85 mg/kg). Statistical analyses provided substantial evidence supporting the assertion that barium content exhibits significant regional variations, with higher concentrations in populations situated in the eastern regions (ST14 to ST17) as opposed to those in the central (ST12–ST13) and southwestern regions of the Iberian Peninsula (ST1 to ST11) (refer to Figure 2). Conversely, the strontium content demonstrated a discernible disparity, with elevated levels detected in both central and eastern peninsular populations relative to those observed in samples analyzed from the southwestern region (see Figure 2).

In the specific case of *A. meridionale*, the concentration of barium was marginally higher than that in *A. macrostachyum*, reaching maximum values of up to 20.54 mg/kg (refer to Table 2, sample 1). Furthermore, the strontium content in *A. meridionale* exhibited a substantial increase compared to *A. macrostachyum*, particularly noteworthy in samples analyzed from Tunisia, where concentrations ranged from 162.15 mg/kg to 438.38 mg/kg.

The semiquantitative analysis of heavy metals (Cr, As, Cd, and Pb) generally indicated low concentrations across all samples. Despite the absence of statistically significant differences among geographical populations, upper limits in some *A. macrostachyum* populations were notably elevated: Cr (27.97 mg/kg), As (13.03 mg/kg), Cd (1.65 mg/kg), and Pb (38.65 mg/kg). Similarly, *A. meridionale* exhibited low concentrations of arsenic, cadmium, and lead. Nevertheless, chromium values in this species surpassed the maximums recorded in *A. macrostachyum*, ranging from 5.93 mg/kg to 33.24 mg/kg.

The results obtained by X-ray diffraction coincided, except for the calcium element, with the crystals and accumulations found in the succulent stems and inflorescences of both *Arthrocnemum* species. Using this technique, the minerals halite, silvite, natroxalate, and glushinkite have been detected in two representative samples of *A. macrostachyum* (Figure 10).

### 2.2. Seeds

The semiquantitative elemental composition analyzed in the seeds of the two *Arthrocnemum* species (*A. macrostachyum* and *A. meridionale*) revealed values of interest for human nutrition (Table 3). Sodium values in seeds were remarkable (3431.19–11,608.57 mg/kg), although higher when pulverization and ICP-MS analysis were performed without discarding the perianth-pericarp pieces attached to the seed (9952.39–25,498.87 mg/kg).

The concentration of magnesium in both seeds and seeds with perianth-pericarp attached showed relative homogeneity across the various samples, ranging from 2421.98 to 5780.67 mg/kg. Similarly, potassium levels were consistently high and uniform across all samples, ranging from 2526.52 to 7509.57 mg/kg, except for the *A. meridionale* sample from Santa Gila (Sardinia, Italy), which displayed an exceptionally high concentration of 16,244.86 mg/kg.

On the other hand, calcium levels exhibited noticeable variations between the populations of *A. macrostachyum* from the Tinto River, ranging from 1196.26 to 2049.41 mg/kg, and those of *A. meridionale*, ranging from 3570.68 to 9717.41 mg/kg. Interestingly, the concentration of calcium increased when the perianth-pericarp was maintained, reaching levels of 4538.10 mg/kg in *A. macrostachyum*.

Micronutrient compositions, specifically iron (Fe), zinc (Zn), boron (B), manganese (Mn), molybdenum (Mo), copper (Cu), and nickel (Ni), exhibited variability across the examined samples. The iron content in seeds of *A. macrostachyum* from the Tinto River was found to be lower when compared to those of *A. meridionale*. For instance, *A. meridionale* sample S6 from Kraten (Tunisia) displayed a significantly higher iron concentration of 781.12 mg/kg. In the case of *A. macrostachyum*, the iron concentration increased when the perianth-pericarp was retained as opposed to isolating the seeds. Zinc, boron, manganese, molybdenum, copper, and nickel exhibited uniform values across most samples, barring a few exceptions. Notably, the zinc content in *A. macrostachyum* seeds from sample S3 (continental salt flats of Cordovilla, Albacete) was significantly higher at 428.68 mg/kg compared to the rest of the samples. Copper concentration was found to be higher in seeds with perianth-pericarp (88.0–243.78 mg/kg) compared to isolated seeds (12.90–31.55 mg/kg).

The investigation of alkaline earth metals, specifically barium and strontium, revealed varying results among different samples. In the seeds of the populations from the Tinto River, the presence of barium was negligible. However, the Tunisian sample of *A. meridionale* showed the highest recorded value of barium (7.46 mg/kg) among all samples. Additionally, seeds with perianth-pericarp demonstrated an increased barium content. As for strontium, variations were observed between different species and populations. Tinto River seeds of *A. macrostachyum* exhibited low strontium levels, Cordovilla seeds showed slightly higher levels, and the two populations of *A. meridionale* displayed significantly higher and more uniform strontium concentrations (approximately 77 mg/kg).

The semiquantitative analysis of heavy metals known to be harmful to organisms, such as chromium (Cr), arsenic (As), cadmium (Cd), and lead (Pb), revealed minimal levels in the seeds. However, in some cases, when the perianth-pericarp was preserved in the seed, the content of certain elements, particularly lead, ranged from 3.88 to 15.26 mg/kg.

Figure 11 presents a longitudinal section of an *A. macrostachyum* seed observed under a scanning electron microscope (SEM). Through energy-dispersive analysis of X-rays, a comprehensive mapping of the seed’s surface was conducted. This section of the seed exhibited a curved embryo and a substantial amount of perisperm towards the inner region, while the cells at the seed’s edge featured papillae, protruding structures. The dispersive analysis mapping revealed that phosphorus and potassium were primarily concentrated in the embryo, sparsely distributed along the seed’s edge. Sodium, on the other hand, was detected exclusively at the edges and interior of the papillae, indicating consistent crystal structures previously detected in seeds that were not yet fully matured (Figure 4C,D).

## 3. Discussion

### 3.1. Vegetative and Floriferous Succulent Stems

This study investigates the elemental composition and adaptive mechanisms of the halophilic species *Arthrocnemum macrostachyum* and *Arthrocnemum meridionale*. These species hold significance in the domains of human nutrition and livestock feed due to their distinct profiles of macronutrients and micronutrients.

#### 3.1.1. Salt Accumulation (Na and K) and Adaptive Mechanism

In halophilic tissues, the accumulation of salts, particularly sodium, occurs within the cell vacuoles of all tissues [42]. The relative quantitative abundance of sodium in *A. macrostachyum* and *A. meridionale* aligns with the general pattern observed in several halophytes of the Salicornioideae subfamily. Sodium exhibits the highest concentration in the photosynthetic succulent stems of Salicornioideae species [6,9,26,37,43,44].

In addition, in the genus *Arthrocnemum*, sodium accumulation primarily occurs in biologically less active tissues and cells such as xylem, vascular bundles of floral stems, water storage tissue cells, and epidermis. The presence of halite (NaCl) and natroxalate (Na_2_C_2_O_4_) in these tissues suggests a regulated biological process. Storing these compounds in inactive forms may serve as a mechanism to exclude toxic ions, particularly sodium, at the cellular level. This adaptive strategy allows these species to thrive in environments with high salt concentrations.

Maintaining osmotic balance at the cellular level in halophilic species involves a competition between sodium and potassium. In contrast to sodium, potassium presence in the form of sylvite (KCl) in both *Arthrocnemum* samples is lower in crystallization. This suggests that potassium may function as a reservoir that can be mobilized when needed to maintain ion balance [9].

#### 3.1.2. Influence of the Tinto River Environment

The primary source of our analyzed samples is the Tinto River, a region with significant implications for the composition of the plants under study. Notably, the Tinto River marshes stand out as one of the most contaminated estuarine systems globally, largely attributed to historical mining activities, particularly large-scale open-pit mining practices dating back to the late 19th century. The geo-biodynamics along the Iberian Pyrite Belt (IPB) further contribute to the pollution of this system [45,46,47,48].

Within this environmental context, the estuary soils in the Tinto River region maintain an average pH of 6.3, enhancing the bioavailability of highly mobile elements. Tidal influences in this area lead to elevated levels of sodium and chlorine. Moreover, this intricate system is marked by high concentrations of iron and sulfur, while crucial macronutrients such as calcium and potassium are scarce [49,50].

This study primarily focuses on the analysis of 11 populations of the species *A. macrostachyum* collected from the Tinto River (designated as ST1 to ST11). These populations consistently exhibit a pattern of element accumulation. The remarkable ability of *A. macrostachyum* to thrive in these extreme ecosystems underscores its adaptive capacity, resulting in high concentrations of macronutrients like potassium (K), magnesium (Mg), and calcium (Ca), even in the nutrient-poor soils characteristic of the Tinto River. This finding emphasizes the potential of *A. macrostachyum* as a viable edible plant.

However, caution is advised, as some authors recommend avoiding the use of the green stems of *Salicornia* species that grow in estuaries with high heavy metal content, such as those in the Tinto and Odiel Rivers [51]. Our ICP-MS analyses on succulent stems and inflorescences of *Arthrocnemum* align with these recommendations, revealing high contents of heavy metals such as Pb and Cr. Notably, samples with elevated values of heavy metals hold promise in the biotechnological field of phytoremediation. *A. macrostachyum*, in particular, has previously been identified as a cadmium hyperaccumulator halophyte [52].

#### 3.1.3. Exploring the Role of Other Macronutrients (Mg, Ca) and Micronutrients

Notably, our analysis reveals higher concentrations of calcium and magnesium in plants from the central region of the Iberian Peninsula. These findings align with the presence of continental salt lagoons formed on substrates of clay, marl, and gypsum, wherein saline efflorescence residues, comprising alkaline chlorides and diverse sulfates (e.g., CaSO_4_.2H_2_0, Na_2_SO_4_, MgSO_4_), accumulate during the summer. These minerals significantly elevate soil concentrations of calcium and magnesium, concurrently increasing pH levels within the neutral to alkaline range (6.8–8.4) [9].

Rufo et al. [9] recently described in the endemic halophyte of the center of the Iberian Peninsula *Sarcocornia carinata* Fuente, Rufo and Sánchez-Mata (Madrid, Toledo, and Ciudad Real), high magnesium contents in its photosynthetic succulent stems (15,558 to 41,735 mg/kg dry weight), very similar to those detected in our study for *A. macrostachyum* collected in the Mancha lagoons of Toledo (ST12), Murcia (ST16), and the two populations of *A. meridionale* from Tunisia (ST18 and ST19). In the succulent stems and inflorescences of the two species of the genus *Arthrocnemum*, the abundance of immobilized magnesium in the form of glushinskite stands out. This mineral has also been detected in other Salicornoideae species such as *Sarcocornia pruinosa*, *S. carinata*, and the cactus *Opuntia ellisiana* [6,9,39].

The process of biomineralization involving calcium oxalate stands as a captivating instance of a biologically regulated mechanism [53]. Notably, calcium oxalate exhibits variable degrees of hydration, manifesting in two distinct crystalline forms: weddellite (CaC_2_O_4_.2H_2_O) and whewellite (CaC_2_O_4_.H_2_O). Within the Salicorniodeae subfamily, only weddellite has been observed [6], a pattern consistent with findings in other taxa like Cactoideae [54].

These compounds are associated with multiple functions, including defense against herbivores, calcium storage, and metal chelation [39]. Intriguingly, certain species within the taxonomic order Caryophyllales exhibit a unique behavior: their crystals decompose during daylight, releasing CO_2_, particularly when the plants experience heightened water stress. Conversely, these crystals reform during the night, effectively serving as a reservoir for CO_2_ provision to the plant. This cycle of decomposition and reformation of calcium oxalate crystals has been documented as a strategy in drought environments [8], underlining the crucial role played by carbon dioxide and water generated during this process.

Regarding micronutrients, iron is one of the most relevant due to the high concentrations detected both in stems and seeds. It is also recognized for the fundamental role it plays in metabolic processes. Iron is an essential micronutrient for nearly all living organisms, with implications for DNA synthesis, respiration, and photosynthesis. Likewise, the study area in the Tinto River, due to its natural characteristics of having water and sediments with high concentrations of iron, has served as a model to study the content, management, and distribution of this important element in plants [40,44,55].

The analysis of TEM images of *A. macrostachyum* succulent stems has enabled the detection of ferritin protein in chloroplasts. The visualization of TEM images of the succulent stems of *A. macrostachyum* analyzed has allowed us to detect the protein ferritin in the stroma of the chloroplast cells, just as it has been detected in *Sarcocornia*, *Salicornia*, *Halogeton*, *Imperata*, and *Panicum* [40]. In this work, we demonstrated through EDX analysis associated with the TEM microscope that the crystalline structures of the chloroplast were mainly composed of iron.

Furthermore, through meticulous analysis of the molecular aggregate diameters and the observed diffraction pattern, our research team has reached a compelling conclusion: the iron present in the dicots under study, including samples of *A. macrostachyum* collected from the Tinto River, is predominantly stored in the form of phytoferritin nuclei. Notably, the detection of ferritin within a significant number of chloroplasts carries exceptional significance, given *Arthrocnemum*’s innate trait of possessing numerous photosynthetic succulent stems. This finding is further reinforced by the high values obtained through ICP-MS analysis.

Regarding micronutrients, manganese plays a crucial role in plant development but can become toxic when present in high concentrations, underscoring the significance of maintaining proper metal homeostasis [56]. Analysis of samples from *A. macrostachyum* and *A. meridionale* revealed promising levels of manganese. For adult men, the recommended daily intake (AI) of manganese is 2.3 mg/day, emphasizing the importance of obtaining an adequate supply of this microelement.

Another outstanding element in the samples analyzed is Zn. Like other Salicornoideae, *Arthrocnemum* species could be considered a source of this micronutrient [57,58,59]. Conversely, the consumption of edible succulent shoots and stems of *A. macrostachyum* and *A. meridionale* is only recommended in soils with low copper content, since in some cases, the copper content in the plant exceeds the maximum tolerable intake level (UL) for adults (10 mg/day), a value based on protection against liver damage as a critically adverse effect [60].

Finally, strontium (Sr), which forms divalent cations, was found in notable concentrations, higher than those of other micronutrients such as Mn or Zn. Strontium is not a relevant element for plant metabolism, but it could cause toxicity, even though it is commonly found in plants in its naturally stable form. Physiological mechanisms for strontium uptake appear to be related to its chemical similarity to other elements, such as calcium. It is known that, in some cases, Sr enters the cell through Ca and K transporters, can move through the plant through the xylem and phloem, and can be stored in plant tissues [9,61].

#### 3.1.4. Application in Food Products

In the realm of halophytes belonging to the Salicornioideae subfamily, recent studies have unveiled intriguing possibilities. These investigations propose the utilization of pulverized succulent stems of these plants as healthier alternatives to traditional salt, with applications explored in various food products such as sauerkraut fermentation, snacks, and bread. The incorporation of these pulverized stems not only results in a significant reduction in sodium content in the final product but also introduces bioactive compounds with antioxidant properties, mineral nutrients, and other beneficial components to the diet [62,63,64].

Halophilic species exhibit substantially higher concentrations of elements compared to glycophytes, which are adapted to low-sodium ecosystems and maintain low sodium levels in their aerial tissues [34]. This unique biological characteristic of halophytes opens avenues for potential applications in livestock feed. Incorporating these species into feed formulations may offer a means to provide animals with the necessary nutrients while meeting their sodium requirements. However, further research is essential to fully explore the feasibility and benefits of using halophilic species like *Arthrocnemum* in livestock nutrition strategies.

Based on ICP-MS analysis, both *Arthrocnemum* species emerge as promising sources of “green salt” for food products. Nevertheless, caution is warranted due to anti-nutritional factors associated with halophyte consumption, particularly concerning elevated sodium and calcium levels. The World Health Organization recommends an Acceptable Daily Intake (ADI) of 2 g of sodium to mitigate the heightened risks of hypertension and cardiovascular disease. Additionally, the formation of oxalate poses a concern, as it can hinder calcium absorption, potentially contributing to the development of insoluble complexes associated with urinary kidney stones.

While dietary guidelines advise limiting oxalate-rich foods, it’s noteworthy that halophytes, including *Arthrocnemum* species, akin to other plant-based foods like beets, are typically consumed in moderation, minimizing potential adverse health effects [65]. Further research and careful consideration of nutritional implications are crucial to fully harnessing the benefits of halophytes in promoting both human and animal health.

### 3.2. Seeds

The data on the elemental content in seeds constitutes a novelty in this study for the two species of the genus *Arthrocnemum*. de la Fuente et al. [6] analyzed the mineral content of seeds in the *Sarcocornia pruinosa* species, providing values for important elements such as Na, Ca, Mg, K, Mn, and Fe; and heavy metals such as As and Pb. Our data on the populations of the species *A. macrostachyum* and *A. meridionale* are close to those described for this halophyte. However, the lead element content is lower in seeds without the perianth-pericarp of *Arthrocnemum* (0–0.34 mg/kg d.w.) than in seeds of *S. pruinosa* from the Tinto River (5.30 mg/kg d.w.). Consequently, we claim in this study the promotion of the use of the seeds coming from *Arthrocnemum* species in food, for example, producing seed bread.

Indeed, its use is a substantial point to consider since it maintains a relative similarity with the seeds of the widely used poppy (*Papaver rhoeas* L.) in bread and cakes [66]: both seeds are black, with a crustaceous testa, small in size, and the respective plants produce a large quantity of them. Likewise, the seeds of *Arthrocnemum*, apart from the beneficial elements that it presents for health, also show a very low heavy metal content, even when they come from plants that grow in toxic soils such as those of the Tinto River.

The values obtained in *Arthrocnemum* seeds allow its comparison with the native species of South America, *Chenopodium quinoa* Willd, it is one of the plants in the Chenopodiaceae family most used worldwide for food [67]. These authors, in the compendium on the nutritional potential of quinoa, refer to values in seeds for the elements calcium (275–1487 mg/kg), copper (10–95 mg/kg), iron (14–167 mg/kg), magnesium (260–502 mg/kg), phosphorus (1400–5300 mg/kg), potassium (6967–14,750 mg/kg), sodium (110–310 mg/kg), and zinc (28–48 mg/kg). The mineral composition values (P, K, Ca, Mg, Fe, Na, and Zn) analyzed in different cultivars of *C. quinoa* by Reguera et al. [68] and Rodríguez et al. [69] also follow the frequently reported concentrations for this species.

Our values in *Arthrocnemum* seeds for the elements K, Ca, Fe, Cu, and Zn are generally slightly higher than those obtained in *C. quinoa*. Additionally, the elemental content of Na (3431–11,608 mg/kg) and Mg (2421–5780 mg/kg) in *Arthrocnemum* seeds is significantly higher than that of quinoa, at least by one order of magnitude.

In the mapping by dispersive energy analysis of X-rays associated with the scanning electron microscope (SEM), it is shown that *Arthrocnemum* seeds accumulate potassium and phosphorus specifically in the embryo. The perisperm is composed of abundant starch [70]. Of special interest is the detection of reduced sodium chloride crystals on the edges and inside the papillae of the *A. macrostachyum* seed. Montserrat-Marti [71] in the genus *Moehringia* L. (Caryophyllaceae) suggested that the seed papillae could be interpreted as an adaptation to drought since an increase in their surface facilitates water retention and therefore favors germination. We suggest that in the genus *Arthrocnemum*, apart from this plausible functionality, we add the following: it is very likely that due to the toxicity that sodium ions can generate in the development and capacity of the embryo, their immobilization and isolation occur in their crystalline form (NaCl) along the edges of the seed and inside the conical papillae or subpapillae.

In addition to the micrometer-sized elements and crystals found in *Arthrocnemum*, for the species *S. pruinosa* collected in the Tinto River, we previously detected abundant iron nanoparticles by transmission electron microscopy (TEM) in longitudinal sections of its seeds [6]. Given the high iron content present in *Arthrocnemum* seeds by the ICP-MS technique, its study under the TEM microscope likely provides data similar to those obtained in *S. pruinosa*.

## 4. Materials and Methods

### 4.1. Studied Material

Whole fresh specimens were collected directly from the field in various territories, in different seasons: spring-summer (flowers) and summer-autumn (fruits and seeds) (Figure 12). Plant material was stored in a refrigerator at 4 °C and in a freezer at −20 °C, for subsequent analysis; some specimens were also dried by pressing, later frozen at −20 °C and finally stored in our collections at the Universidad Autónoma de Madrid (UAM) or deposited in the Herbarium of Pharmacy of the Universidad Complutense de Madrid (MAF). All the samples collected and/or analyzed in this study are included in Appendix A.

Figure 13 displays a representative example of the key characteristics for recognizing the genus *Arthrocnemum*, including succulent stems, inflorescences, perianth, and seeds.

### 4.2. Inductively Coupled Plasma Mass Spectrometry (ICP-MS)

The elemental content was analyzed and quantified by inductively coupled plasma mass spectrometry (ICP-MS), a highly sensitive technique for the determination of multiple elements. Plant samples were analyzed using the protocol described by Zuluaga et al. [72], which optimizes the digestion process of plant matter and allows the optimal recovery of a large number of elements in a single semi-quantitative analysis. After cleaning, a plant and soil sample of approximately 500 mg from each locality was weighed in a Teflon tube for acid digestion in a mixture of 8 mL of 65% HNO_3_ and 2 mL of 30% H_2_O_2_ inside an MLS Ethos 1600 URM Milestone high-pressure microwave digester. The volume of each sample was subsequently adjusted to 25 mL with deionized water.

Aliquots of the solutions obtained from plant and seed samples were analyzed for Na, Mg, K, Ca, Fe, Zn, B, Mn, Mo, Cu, Ni, Ba, Sr, Cr, As, Cd, and Pb by ICP-MS using an ICP ELAN-600 PE Sciex Instrument (Toronto, ON, Canada). This technique determined the concentrations in mg/kg, based on the dry weight of the digested sample. The ICP-MS technique has an inherent error of 15%.

The plant materials used were succulent stems and inflorescences (measured in triplicate) and individualized seeds and/or seeds with attached perianth-pericarp (in these cases, one measurement was made per sample). The statistical treatment of the data obtained was carried out using the IBM SPSS Statistics 26 program. The differences between the mean values analyzed for each of the variables SW (southwest of the Iberian Peninsula, Tinto River), C (Center of the Iberian Peninsula), and E (East of the Iberian Peninsula) were tested with a multivariate factorial ANOVA. The cases responsible for significant main effects were detected using the Bonferroni post hoc test, with a level of significance of *p* < 0.05 (Figure 1 and Figure 2).

### 4.3. X-ray Diffraction Analysis

X-ray diffraction patterns for each analyzed sample were obtained using a Siemens-D5000 (Siemens AG, Karlsruhe, Germany), a diffractometer with Cu Kα (8.04 keV) radiation, and an SOL-X 249 detector provided by Bruker (Billerica, MA, USA). The samples were analyzed using the unoriented powder method. The components were identified using the patterns registered in the crystallographic database PDF-4 of the ICCD (International Centre for Diffraction Data). The diffractograms were carried out in the Servicio Interdepartmental de Investigación from the Universidad Autónoma de Madrid (UAM, Spain).

For plants, succulent and woody stems were separated, cleaned, dried, and powdered using an IKA A11 basic instrument. Samples were analyzed using an X’Pert PRO Theta/2Theta (Almelo, The Netherlands) analyzer with a graphite monochromator for Cu K-alpha-1 wavelength (1.5406 Å) and an X’Celerator fast detector. Identification was carried out using the HighScore Plus software created by Panalytical Plus and the ICDD PDF-4+ Full File database. Analyses and identification were conducted in the SIDI-UAM.

### 4.4. Scanning Electron Microscopy (SEM-EDX)

The plant samples collected were analyzed by SEM and an energy-dispersive X-ray analyzer (EDX). In this study, we followed the methodology for the analysis of elements and localization of elements in plant material described by Rodríguez et al. [40]. The organs and tissues analyzed were succulent stems. Dry samples were cut into cross and longitudinal sections; these were then mounted onto conductive graphite stubs and sputters and coated in gold in a BIORAD SC 502 apparatus. The preparations were studied with a Hitachi S-3000 N (Tokyo, Japan) SEM coupled with an INCAx-sight and Si-Li detector (Oxford, UK). An acceleration voltage of 20 kV and a working distance of 15 mm were used in the analyses that were performed at room temperature.

### 4.5. Transmission Electronic Microscopy (TEM)

Sections of plant tissues of approximately 1 mm^3^ were fixed in the Electron Microscopy Service of the Centro de Biología Molecular Severo Ochoa (CBMSO) with the method described in Fuente et al. [6]. Samples were observed in the same service with a JEM-1010 transmission electron microscope and in the Centro Nacional de Microscopía Electrónica de Madrid with a JEM 2000FX electron microscope (JEOL, Tokyo, Japan) operated at 200 kV, coupled with an energy dispersive X-ray microanalysis instrument LINK ISIS 300 (Oxford Instruments, Oxford, UK).

## 5. Conclusions

In conclusion, this study provides valuable insights into the elemental composition of *Arthrocnemum macrostachyum* and *Arthrocnemum meridionale*, shedding light on their unique strategies for coping with high salt concentrations in their habitats. The findings may have important implications in various fields, including human nutrition and the understanding of plant adaptations to extreme environments. Moreover, the exquisite blend of highly substantial macronutrients and micronutrients, coupled with the conspicuously low levels of heavy metal presence, positions *Arthrocnemum* seeds as an exceptional nutritional source for prospective utilization.

## Figures and Tables

**Figure 1 plants-13-00496-f001:**
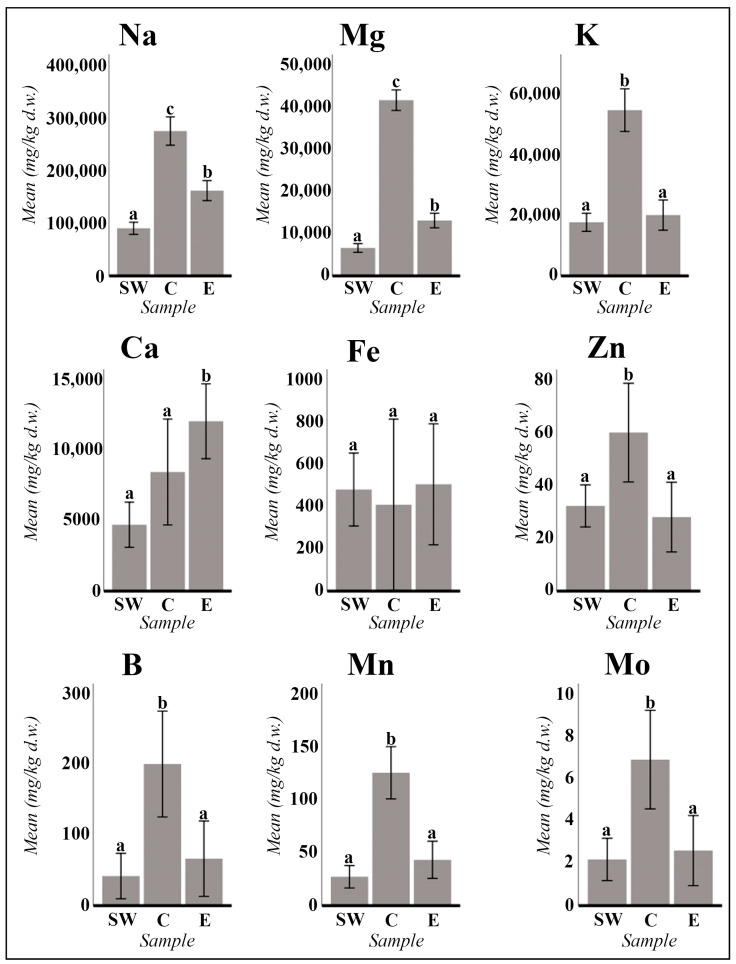
*A. macrostachyum* mean values in mg/kg d.w. (dry weight) of the main macronutrients (Na, K, Mg, and Ca) and micronutrients (Fe, Zn, B, Mn, and Mo). The mean values of the populations of the southwest of the Iberian Peninsula (SW; samples ST1 to ST11; n = 33), of the Center of the Iberian Peninsula (C; samples ST12 to ST13; n = 6), and of the East of the Iberian Peninsula (E; samples ST14 to ST17; n = 9) are represented. Error bars are at the 95% confidence interval. Different lower-case letters indicate significant differences between means (*p* < 0.05).

**Figure 2 plants-13-00496-f002:**
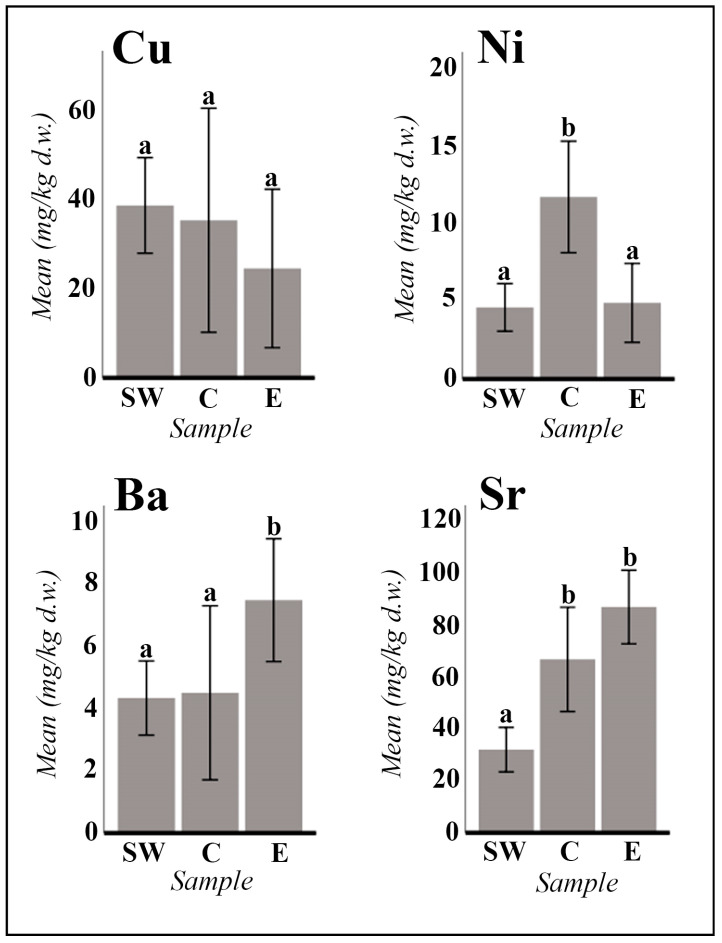
*A. macrostachyum* mean values in mg/kg d.w. (dry weight) of the main micronutrients (Cu and Ni) and alkaline earth metals (Ba and Sr). The mean values of the populations of the southwest of the Iberian Peninsula (SW; samples ST1 to ST11; n = 33), of the Center of the Iberian Peninsula (C; samples ST12 to ST13; n = 6), and of the East of the Iberian Peninsula (E; samples ST14 to ST17; n = 9) are represented. Error bars are at the 95% confidence interval. Different lowercase letters indicate significant differences between means (*p* < 0.05).

**Figure 3 plants-13-00496-f003:**
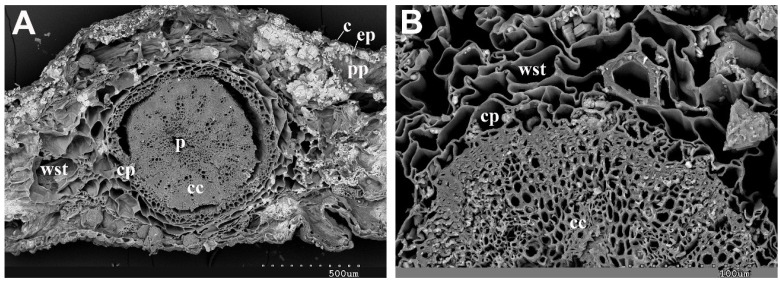
Representative micrographs of *Arthrocnemum* cells and tissues. Transversal section of succulent stem obtained by scanning electron microscopy: (**A**) *A. macrostachyum* (Lobos Island, Fuerteventura). (**B**) *A. meridionale* (Sfax, Tunisia). Symbology: c (cuticle); ep (epidermis); pp (palisade parenchyma cells); wst (water storage tissue); cp (cortical parenchyma cells); cc (central cylinder); p (pith).

**Figure 4 plants-13-00496-f004:**
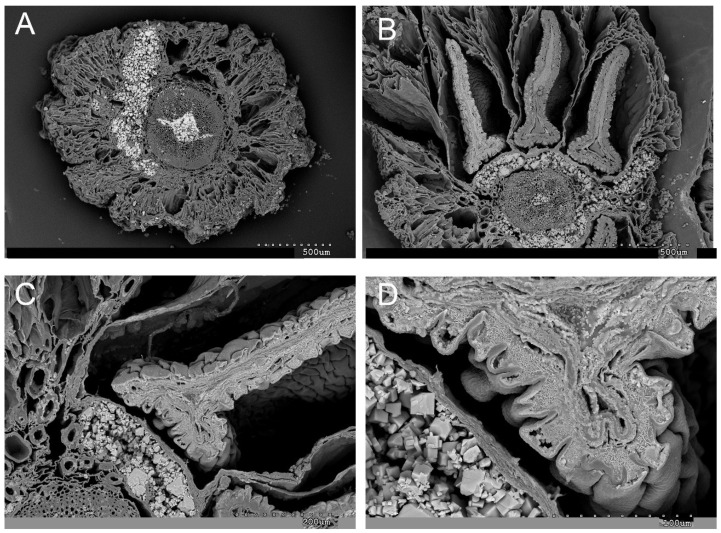
Representative SEM micrographs of transversal sections of *A. macrostachyum* (San Juan del Puerto, Huelva). (**A**) Cross section of a succulent stem where the pith, the cells of the cortical parenchyma, and sections of the water storage tissue are completely collapsed by the biomineralization process. (**B**) Cross section of inflorescence with the three flowers per cyme. (**C**) Detail of a flower with ovary and seed in the maturation phase. (**D**) Na and Cl crystals in the parenchymatic tissues of the flower and inside the cells and tissues of developing seeds.

**Figure 5 plants-13-00496-f005:**
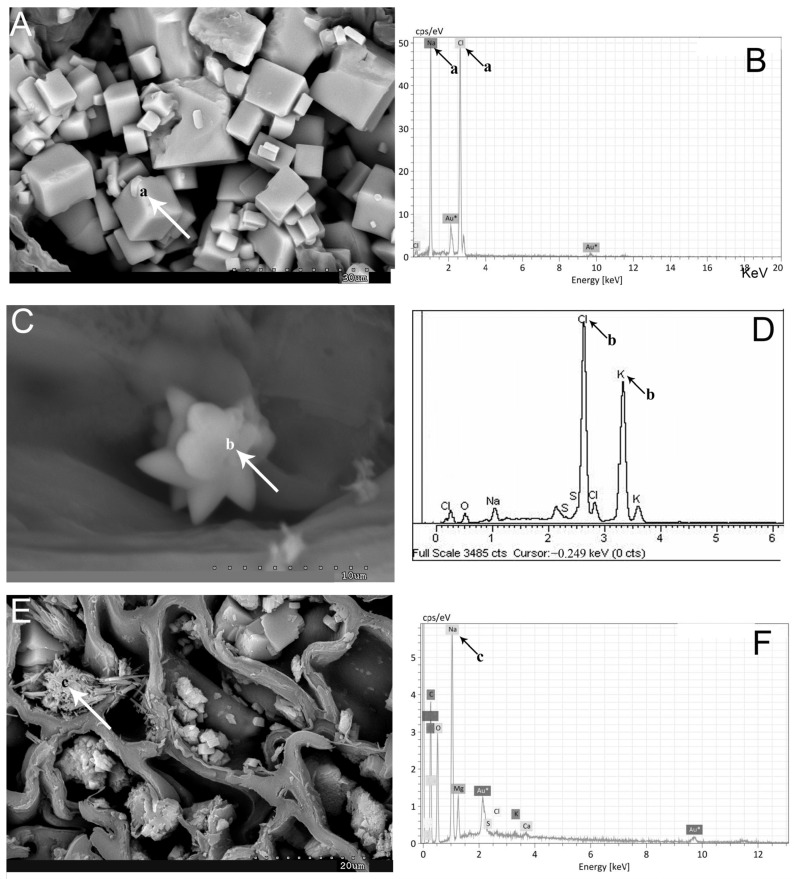
Representative SEM images of Na and K biominerals in *A. macrostachyum* and *A. meridionale*: (**A**) Cubic NaCl crystals in different cells and tissues (Villasequilla de Yepes, Toledo). (**C**) Stellate K and Cl crystals (San Juan del Puerto, Huelva). (**E**) Rectangular plates and fibers of Na, C, O, and S (Sfax, Tunisia). Spectra by energy dispersive X-ray analysis (**B**,**D**,**F**). Arrows and lowercase letters indicate EDX analysis.

**Figure 6 plants-13-00496-f006:**
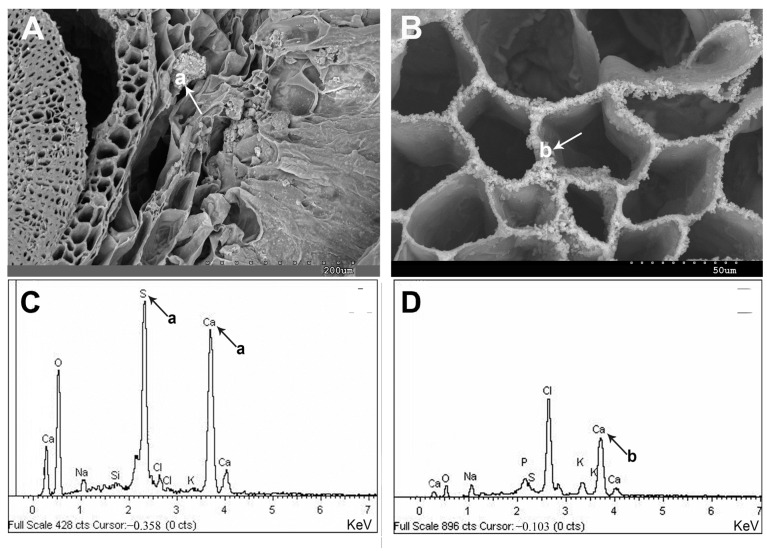
Representative SEM images of calcium accumulations in *A. macrostachyum* and *A. meridionale*: (**A**) Ca and S in the water storage tissue (Vendicari, Sicily), (**B**) Ca biominerals covering the xylem cell walls (La Rábida, Huelva), (**C**,**D**) Calcium spectra by energy dispersive X-ray analysis. Black and white arrows and lowercase letters correspond to Ca minerals.

**Figure 7 plants-13-00496-f007:**
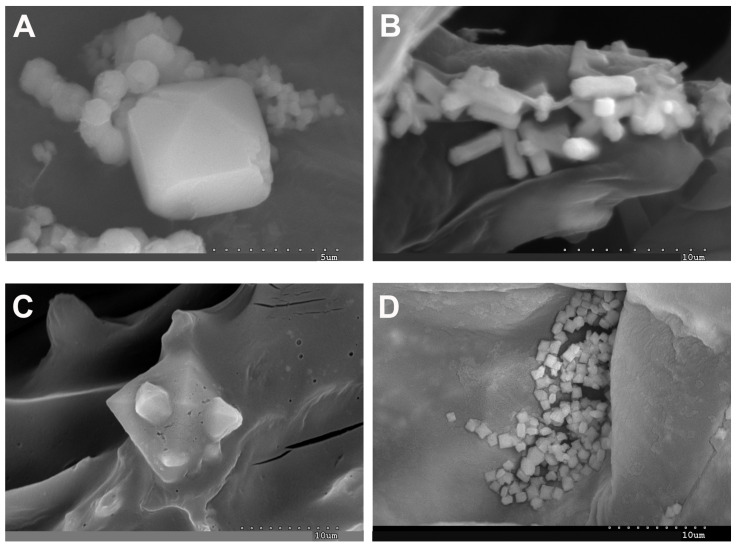
Representative SEM images of Ca oxalate crystals (weddellite) in *A. macrostachyum*: (**A**) Prismatic crystals (San Juan del Puerto, Huelva), (**B**) Rectangular crystals (Villasequilla de Yepes, Toledo), (**C**) Polyhedral structures (Torreblanca, Castellón), (**D**) Cubes (Rábida, Huelva).

**Figure 8 plants-13-00496-f008:**
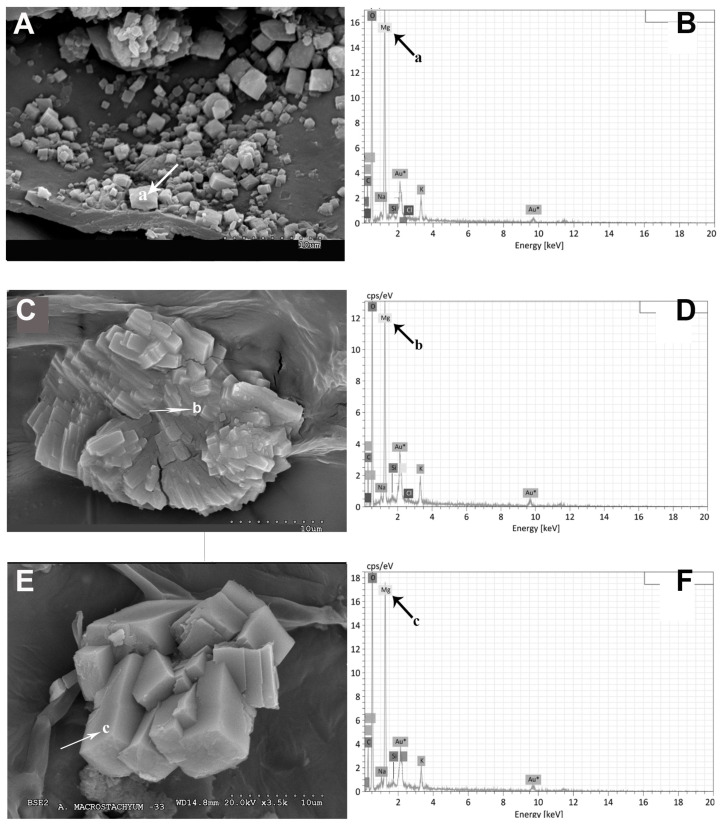
Representative SEM images of Mg oxalate biomineral (glushinskite) in the form of cubes and druses in cell walls: (**A**) *A. macrostachyum* (San Juan del Puerto, Huelva), (**C**) *A. macrostachyum* (Santa Pola, Alicante), (**E**) *A. macrostachyum* (San Juan del Puerto, Huelva), Black and white arrows and lowercase letters indicate energy-dispersive X-ray analysis (**B**,**D**,**F**).

**Figure 9 plants-13-00496-f009:**
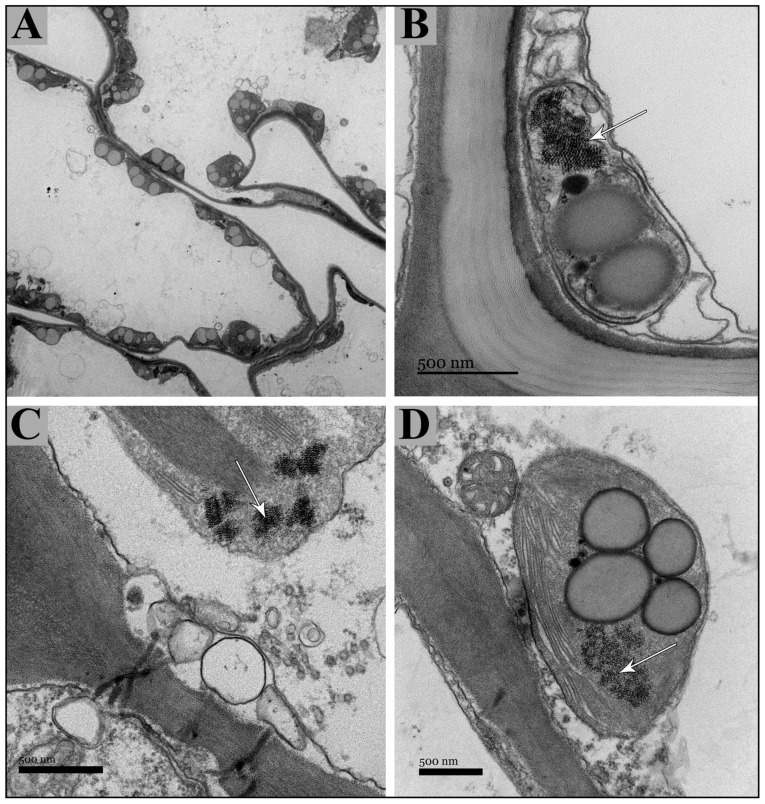
Representative images of chloroplasts with ferritin nuclei in *Arthrocnemum macrostachyum*. (**A**) Chloroplasts surrounding cortical parenchyma cells, (**B**–**D**) Chloroplasts with ferritin in the stroma. White arrows indicate ferritin nuclei, (**A**,**C**,**D**) (Estuary of the Tinto River, Huelva), (**B**) (Rábida, Huelva).

**Figure 10 plants-13-00496-f010:**
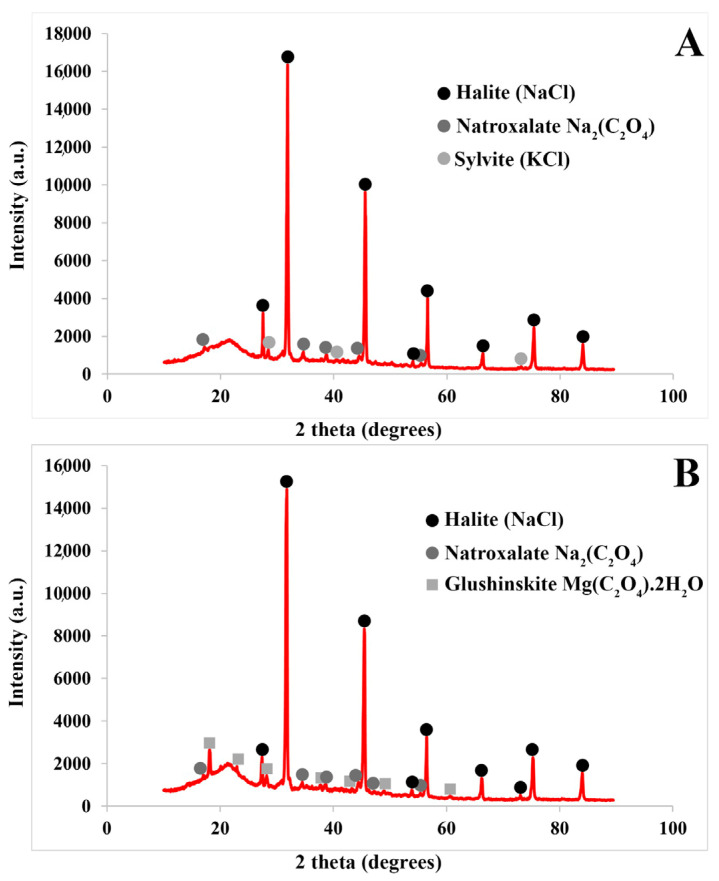
Representative X-ray diffraction spectra of succulent stems of *A. macrostachyum*. (**A**) Halite, sylvite, and natroxalate, (**B**) Halite, natroxalate, and glushinskite.

**Figure 11 plants-13-00496-f011:**
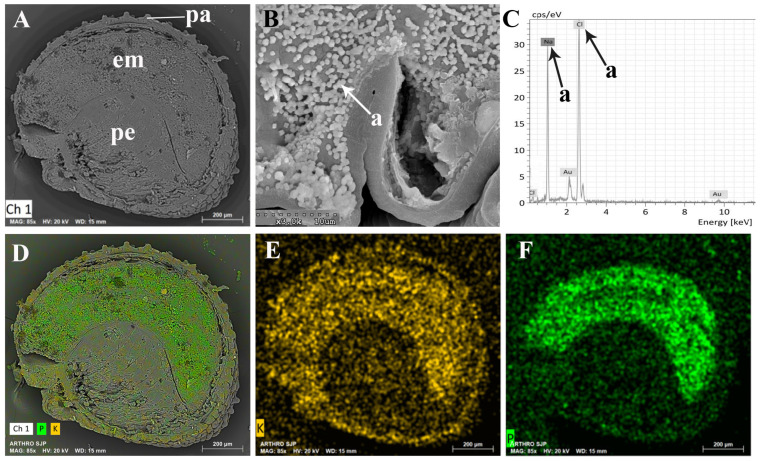
Representative SEM images (**A**,**B**) and SEM-EDX spectra (**C**–**F**) in *A. macrostachyum* (San Juan del Puerto, Huelva). (**A**) Longitudinal section of the seed, (**B**) Detail of NaCl microcrystals inside of the epidermal papillae, (**C**) Spectra by energy dispersive X-ray analysis. Black and white arrows and lower-case letters correspond to Cl and Na elements, (**D**–**F**) Seed longitudinal section mapping (P and K were the elements detected). Symbology: pa (epidermal papillae); em (embryo); pe perisperm. *Arthrocnemum macrostachyum*.

**Figure 12 plants-13-00496-f012:**
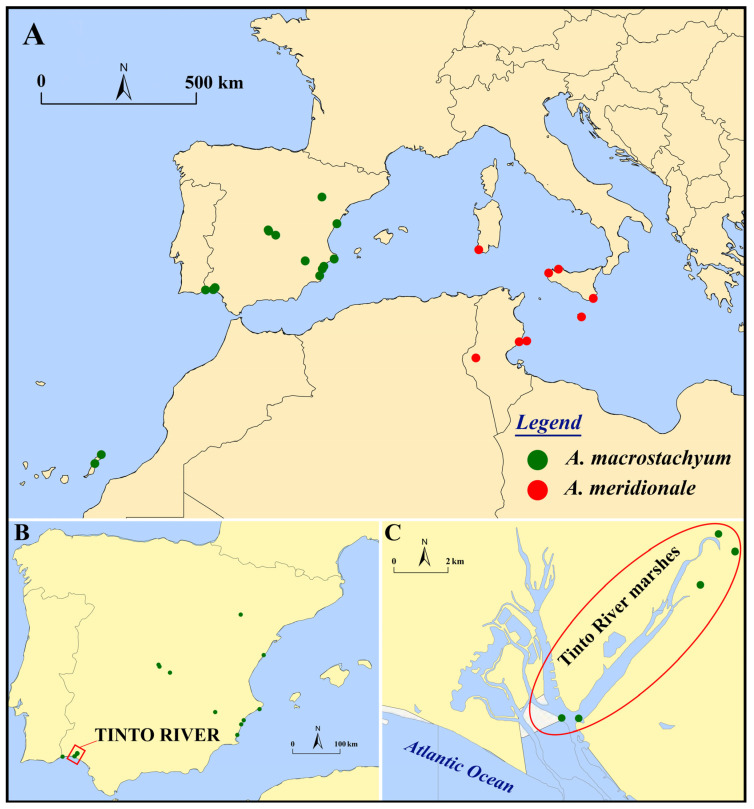
(**A**) Map showing the distribution of *Arthrocnemum macrostachyum* (green circles) and *Arthrocnemum meridionale* (red circles) samples. (**B**) Main area of study in the southwestern of the Iberian Peninsula (Tinto River, Huelva, Spain). (**C**) Tinto River marshes samples. Refer to Appendix A for sample information and analyses.

**Figure 13 plants-13-00496-f013:**
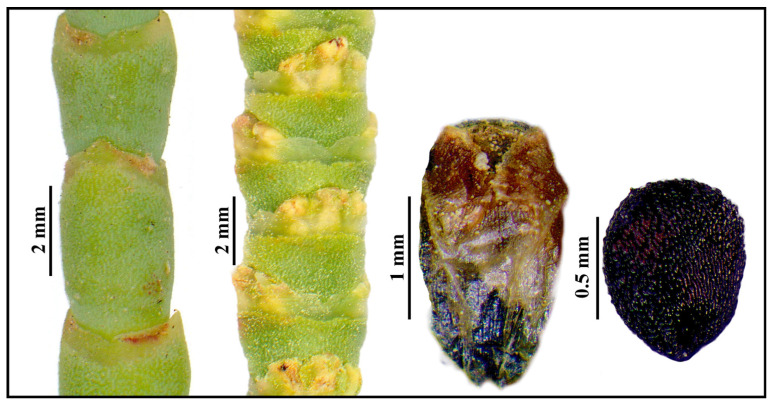
Morphological details of the genus *Arthrocnemum* based on an *A. macrostachyum* Tinto River saltmarsh sample (binocular stereomicroscope). From left to right: succulent vegetative stems; flowering stems in anthesis; perianth structure; black seed.

**Table 1 plants-13-00496-t001:** Elemental composition of succulent stems of *A. macrostachyum*. Data are expressed in mg/kg d.w. (dry weight). The mean values of the populations of the southwest of the Iberian Peninsula (SW; samples ST1 to ST11; n = 33), the Center of the Iberian Peninsula (C; samples ST12 to ST13; n = 6), and the East of the Iberian Peninsula (E; samples ST14 to ST17; n = 9) are represented. Symbology: Mean (M); n = 3. Full data (replicates and standard deviation) and the sample reference list are included in Appendix A (ST1–ST17). In bold is the highest value, and in italics is the lowest value per column.

ID	Na	Mg	K	Ca	Fe	Zn	B	Mn	Mo	Cu	Ni	Ba	Sr	Cr	As	Cd	Pb
ST1 (SW)	91,250.06	6286.93	23,498.67	3002.96	247.79	19.14	23.15	18.94	0.98	9.99	6.95	2.22	*13.62*	14.63	1.06	0.15	0.78
ST 2 (SW)	120,166.30	7088.59	33,717.76	5199.45	*77.78*	35.22	32.50	7.89	1.29	*5.92*	0.87	*0.77*	37.76	6.90	*0.00*	0.20	0.24
ST 3 (SW)	66,367.06	9042.43	15,197.18	8104.79	449.23	27.61	61.64	13.37	*0.55*	28.61	3.82	4.33	64.53	7.35	2.29	0.34	3.21
ST 4 (SW)	122,968.31	6328.36	27,103.31	6531.38	243.42	*16.22*	33.65	24.13	1.00	11.94	*0.37*	4.86	24.02	*0.73*	1.28	0.40	2.35
ST 5 (SW)	114,158.07	7750.63	*6150.54*	8162.98	670.65	37.34	47.95	20.38	1.46	34.64	**14.71**	8.76	27.60	**27.97**	3.59	**1.65**	7.59
ST 6 (SW)	77,774.43	6048.84	16,654.74	2884.32	209.29	24.18	35.07	7.58	1.76	35.51	1.09	2.66	35.21	3.67	0.53	0.06	1.60
ST 7 (SW)	58,089.68	6031.30	10,351.36	3458.96	317.03	19.28	45.98	73.68	1.33	52.85	4.57	2.76	38.97	9.27	1.09	0.08	3.46
ST 8 (SW)	*57,747.82*	6358.98	11,933.12	4346.96	266.93	23.29	36.93	23.44	1.65	50.01	2.53	2.54	28.31	4.26	0.27	0.05	3.52
ST 9 (SW)	76,702.87	7757.82	13,588.86	4265.36	**2135.60**	**107.03**	76.47	69.84	**11.58**	**72.38**	5.78	8.52	41.19	12.27	**13.03**	0.13	**38.65**
ST 10 (SW)	146,528.62	5398.36	21,465.90	4029.47	358.47	19.66	39.26	34.00	1.15	54.03	8.04	5.16	19.32	17.86	0.90	0.58	1.87
ST 11 (SW)	62,742.67	*4739.91*	16,343.87	*1829.82*	319.68	26.10	27.30	*6.25*	1.24	71.40	1.93	5.11	21.64	4.29	1.11	*0.00*	4.30
ST 12 (C)	**311,083.18**	41,491.40	45,374.40	5122.27	418.10	79.74	**397.30**	**173.69**	10.34	45.56	14.33	2.29	39.86	9.11	0.41	0.24	*0.21*
ST 13 (C)	239,613.72	**41,993.52**	**64,870.74**	11,817.14	400.32	40.58	4.76	77.80	3.48	25.43	9.11	6.71	93.72	10.54	0.28	0.98	0.45
ST 14 (E)	142,924.73	13,760.08	15,337.49	8136.56	485.43	18.86	266.07	36.06	4.56	6.06	13.56	5.41	91.41	26.61	0.28	0.27	0.72
ST 15 (E)	193,966.51	7110.62	28,936.80	7263.91	400.82	41.50	*0.00*	13.82	1.46	15.79	2.89	7.31	57.10	1.10	0.31	0.16	1.01
ST 16 (E)	130,930.10	21,258.66	15,266.57	**26,030.18**	1004.92	34.76	*0.00*	97.61	2.50	70.60	2.04	**15.32**	**146.85**	2.54	0.68	0.46	3.46
ST 17 (E)	180,198.04	10,452.45	21,346.67	6911.34	133.91	17.01	*0.00*	25.30	1.87	6.08	1.10	1.89	52.67	0.99	0.13	0.06	0.23

**Table 2 plants-13-00496-t002:** Elemental composition of succulent stems of *A. meridionale*. Data are expressed in mg/kg d.w. (dry weight). Symbology: Mean (M); n = 3. Full data (replicates and standard deviation) and the sample reference list are included in Appendix A (ST18–ST20). In bold is the highest value, and in italics is the lowest value per column.

ID	Na	Mg	K	Ca	Fe	Zn	B	Mn	Mo	Cu	Ni	Ba	Sr	Cr	As	Cd	Pb
ST18	95,912.62	20,554.81	13,271.65	**25,407.54**	**1833.34**	*19.10*	0.00	**89.51**	3.75	5.20	**15.46**	**20.54**	**438.38**	**33.24**	**0.63**	0.70	**1.17**
ST19	*49,275.46*	**21,597.93**	*4972.64*	11,762.08	543.45	**61.41**	0.00	35.70	*1.28*	**10.55**	8.19	11.25	162.15	15.89	0.35	**0.90**	1.01
ST20	**218,830.71**	*13,095.79*	**36,941.86**	*10,772.48*	*247.93*	44.58	0.00	*14.55*	**3.86**	*4.60*	*4.99*	*2.93*	*66.22*	*5.93*	*0.25*	*0.12*	*0.31*

**Table 3 plants-13-00496-t003:** Elemental composition of succulent seeds and/or seeds with perianth and pericarp of *A. macrostachyum* and *A. meridionale*. Data are expressed in mg/kg d.w. (dry weight). The sample reference list is included in Appendix A (S1–S7). In bold is the highest value, and in italics is the lowest value per column.

ID	Especie	Muestra	Na	Mg	K	Ca	Fe	Zn	B	Mn	Mo	Cu	Ni	Ba	Sr	Cr	As	Cd	Pb
S1	*A. macrostachyum*	Seeds	7542.48	*2421.98*	6561.01	1196.26	*93.43*	65.60	31.24	*18.86*	0.04	*12.90*	*0.00*	0.17	*6.84*	*0.00*	*0.00*	*0.00*	*0.00*
S2	*A. macrostachyum*	Seeds	3840.42	2730.72	7509.57	1722.64	173.54	47.13	49.29	**83.76**	0.45	13.56	0.47	0.31	8.64	0.45	0.29	*0.00*	0.10
S3	*A. macrostachyum*	Seeds	*3431.19*	**5780.67**	7314.83	2049.41	100.85	**428.68**	74.37	30.85	0.75	13.59	2.02	*0.00*	24.12	*0.00*	0.23	*0.00*	0.19
S4	*A. macrostachyum*	Seed with pericarp	**25,498.87**	3510.64	7421.10	4538.10	400.77	*35.62*	*0.00*	50.32	0.53	88.00	**6.73**	7.34	184.45	1.37	1.91	**0.25**	3.88
S5	*A. macrostachyum*	Seed with pericarp	9952.39	5320.39	*2526.52*	*85.00*	757.14	73.77	**103.42**	29.99	**2.66**	**243.78**	1.22	**7.76**	**218.31**	**1.81**	**2.79**	0.20	**15.26**
S6	*A. meridionale*	Seeds	4305.88	4071.00	4727.96	**9717.41**	**781.12**	62.34	45.08	55.54	*0.00*	13.40	2.63	7.46	77.20	*0.00*	*0.00*	0.01	*0.00*
S7	*A. meridionale*	Seeds	11,608.57	3943.86	**16,244.86**	3570.68	236.02	95.53	66.38	65.69	0.51	31.55	2.37	1.43	76.51	0.46	*0.00*	*0.00*	0.34

## Data Availability

Data are contained within the article and Appendix A.

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
