# Peer review of "Arthrocnemum Moq.: Unlocking Opportunities for Biosaline Agriculture and Improved Human Nutrition"

_plants, 2024, doi:10.3390/plants13040496_

Round 1
Reviewer 1 Report
Comments and Suggestions for Authors
The paper provides new and valuable data related to anatomy, phyisiology, ecolog and practical use of two studied species from the genus Arthrocnemum: A. macrostachyum and A. meridionale. The field work and sampling design were done properly. The analyses of the elemental content, macronutrients and micronutrients brings relevant numbers about concentration of these elements in plant tissue. Additionaly, x-ray diffraction analyses and use of electronic microscopy showed an exceptioanal figures that are very instructive in presenting the obtained results. One of the achievements of this research is that Arthrocnemum seeds accumulate K and P specifically in the embryo, which is visualy presented in high quality photos. The authors successfully connect environmental pollution with heavy metals by previous mining activities in the Tinto River marshes as a very extreme environment. All results are presented and compared between Arthrocnemum population at sites in Spain and Tunisia. Discussion is well arranged and contain lot of comparison and links to other researches.
The specific suggestions and recommendations are following:
Row 416-419: Delete text that belong to description in Fig. 9, probably mistake.; Row 530: "the protein ferritin in the stroma of the chloroplast cells", change to: "the protein ferritin in the stroma of the chloroplast in cells";
Row 618: Chapter Material and Methods, it is good to provide background about geographical locations of the sampling sites (geographical map, table witlh list of samplin sites and coordinates ) so the reader can be familiar about the surveyed areas in Spain and Tunisia; Row 710: Chapter References, check the use of journal abbreviation in the names of all journals listed in the reference list (references number 17, 22 26, 45, 46, 47, 48). For reference number 73 the name of the journal (abbreviation) is missing !
The English Language in the submitted manuscript is of good quality.
Author Response
REVIEWER 1 (in black our response)
Row 416-419: Delete text that belong to description in Fig. 9, probably mistake.;
We have deleted the text, it was a mistake.
Row 530: "the protein ferritin in the stroma of the chloroplast cells", change to: "the protein ferritin in the stroma of the chloroplast in cells";
We have changed it by The analysis of TEM images of A. macrostachyum succulent stems has enabled the detection of ferritin protein in chloroplasts, …
Row 618: Chapter Material and Methods, it is good to provide background about geographical locations of the sampling sites (geographical map, table witlh list of samplin sites and coordinates ) so the reader can be familiar about the surveyed areas in Spain and Tunisia;
We have incorporated a new map of the samples studied (Figure 12).
Row 710: Chapter References, check the use of journal abbreviation in the names of all journals listed in the reference list (references number 17, 22 26, 45, 46, 47, 48). For reference number 73 the name of the journal (abbreviation) is missing !
We have corrected the mentioned abbreviations as well as some of them. We also have added the missing journal name (Lagascalia) in cite 73.
Reviewer 2 Report
Comments and Suggestions for Authors
In this manuscript, the authors conducted a detailed investigation of the inhabit regions and element accumulation patterns of Arthrocnemum halophytes, that inhabit the coastal and inland areas of the Iberian Peninsula, and contains very interesting findings. Inaddtion, the manusctipt also included interesting discussions related to these results. However, I would like to sugest that authors should give more thought to the structure of the manucript to improve readability.
1. Map
Although the area Iberian peninsula may be familiar to authors, I think it will be easier for readers to understand if there is a map, so I strongly recommend to display the map in the manusctipt.
I also thought it would be better if you show some photos of plants at the collection sites in the manuscript.
2. Decimal point and thousand separator
Regarding the decimal point, there is a mix of commas and periods, so I recommend to organize and uniform them through the manusctipt.
Also, since numbers with a large number of digits were seen in the texts, figures, and tables , it is strongly recommended to display a thousand separator.
I think it would be better to standardize the notation to '49,275.46 mg/kg' in L172 of your manuscript.
3. Element notation
The notation of elements in the text is a mixture of element symbols (Na, Fe, etc.) and genral names (sodium, iron, etc.), but I don't know how they are used respectively.
(The reviewer is not a native speaker, so if there is no problem with mixed notation, you can leave it as is.)
4. L128-131
What do Samples 1 to 3 mean?
5. Table
It is recommended that numbers in tables be displayed using the same decimal point and thousand separator as in the text.
Also, readability can be improved by adding the sampled region (SW, C, E) to the ID row of the table.
On the other hand, you can tell that the numbers in the table are mean by writing them once in regend, (M) are not required.
Also, if there is a graph notation, there is also a supplementary table, so I don't think the table in the main text is necessary.
6. Figure 1 & 2
I did not understand why this diagram is divided into two parts.
The lack of graphs for heavy metals seems to be an attempt to obscure it somehow.
Also, if there is no significant difference, it is preferable to add "a" to all bars.
In addition, although only the stem data of A. macrostachyum is shown as a graph, readability will be improved if the stem data of A. meridionale and seed data are also graphed.
(And no longer needs a table.)
Furthermore, I thought 'samples 1 to 11' in regend probably corresponds to 'samples ST1 to ST11' in Table 1, so please match the numbers in the table and figure.
(Same for L233, L235, L240, L286, L305, L308, L310, L311, L337, L381)
7.Figure 4
As with other SEM image figures, it is desirable to have notes the structure and so on explained in the text, in the figure.
8. Discussion section
The descriptions were very large and it were difficult to understand which data each description refers to, so we strongly recommend that you include citations to figures and tables as well as sample numbers.
Also, readability of the manusctipt will be improved if the notation will be more concise, such as by avoiding duplication with the Results section.
9. Materials and Methods section
The method of statistical analysis should be added.
10. Location of Conclusion section
The Conclusion clause should be locataed immediately after the Discussion section, not after Materials and Methods section.
Author Response
REVIEWER 2 (in black our response)
- Map
Although the area Iberian peninsula may be familiar to authors, I think it will be easier for readers to understand if there is a map, so I strongly recommend to display the map in the manusctipt.
We have incorporated a new map of the samples studied (Figure 12).
I also thought it would be better if you show some photos of plants at the collection sites in the manuscript.
We have incorporated a new figure of a representative image showing Arthrocnemum main recognizable characteristics: Figure 13. Morphological details of the genus Arthrocnemum based on A. macrostachyum Tinto river saltmarsh sample (binocular stereomicroscope). From left to right: succulent vegetative stems; flowering stems in anthesis; perianth structure; black seed.
- Decimal point and thousand separator
Regarding the decimal point, there is a mix of commas and periods, so I recommend to oganize and uniform them through the manusctipt.
Also, since numbers with a large number of digits were seen in the texts, figures, and tables , it is strongly recommended to display a thousand separator.
We have changed the decimal point in figueres, tables and text, periods mean decimals and commas mean thousand separators.
I think it would be better to standardize the notation to '49,275.46 mg/kg' in L172 of your manuscript.
We have changed the text part following this notation
- Element notation
The notation of elements in the text is a mixture of element symbols (Na, Fe, etc.) and genral names (sodium, iron, etc.), but I don't know how they are used respectively. (The reviewer is not a native speaker, so if there is no problem with mixed notation, you can leave it as is.)
No rules about the notation of elements in Plants Journal, we think the paper might sound less repetitive when used in a mixed notation.
- L128-131
What do Samples 1 to 3 mean?
We have changed Sample 1 to 3 to Sample ST18 to ST20, the ones that correspond to the nomenclature used in Table S1 of Supplementary Material.
- Table
It is recommended that numbers in tables be displayed using the same decimal point and thousand separators as in the text.
We have changed the decimal point in tables and text, periods mean decimals and commas mean a thousand separators, in all tables and text.
Also, readability can be improved by adding the sampled region (SW, C, E) to the ID row of the table.
We have added a new column with SW, C, E in the ID section.
On the other hand, you can tell that the numbers in the table are mean by writing them once in regend, (M) are not required.
We have removed the letter (M) in the table
Also, if there is a graph notation, there is a supplementary table, so I don't think the table in the main text is necessary.
We appreciate the suggestion to exclude the table. However, we believe it is beneficial to keep the Mean value in the main text in a condensed way for those who need to reference specific values with decimals. We think it complements the visual figure. Thank you for your input.
- Figure 1 & 2
I did not understand why this diagram is divided into two parts.
We attempted to include all macronutrients, micronutrients, and alkaline earth metals in a single graph. However, our attempts to do so were unsuccessful and resulted in less clarity. Separating them into two different graphs would require reducing the size of each graph to a minimum. Thank you for understanding the limitations we faced in presenting the data effectively.
The lack of graphs for heavy metals seems to be an attempt to obscure it somehow.
In the case of heavy metals, we did not provide graphs because the values are very small but highly variable for some of them (such as Pb and As). The error bars exceed the mean values and appear below the X-axis. Therefore, we believe it is appropriate to maintain the raw table to display the actual data for heavy metals. Additionally, including graphs would increase the number of figures, and they may not be visually clear compared to the other elements. The values are low but highly variable, and we believe this is adequately reflected in the table and the results commentary from our perspective.
Also, if there is no significant difference, it is preferable to add "a" to all bars.
We have added “a” in all the bars with no significant differences.
In addition, although only the stem data of A. macrostachyum is shown as a graph, readability will be improved if the stem data of A. meridionale and seed data are also graphed. (And no longer needs a table.)
We believe that including stem data in a graph on A. meridionale may not make more visual the manuscript on these data. As mentioned in the text, the samples exhibit idiosyncrasies, resulting in the absence of clear patterns. Therefore, we propose presenting the raw data in a table. Regarding the seeds, there is only one measurement per sample: error bars cannot be included and the comparison between different geographic packages is not deemed crucial. The importance of the seed data is the valuable seed content in the genus Arthrocnemum and the absence of heavy metals in separated seeds. Hence, we consider it appropriate to present this information in a table rather than a graph. We sincerely appreciate your suggestions and the effort put into assessing our full manuscript.
Furthermore, I thought 'samples 1 to 11' in regend probably corresponds to 'samples ST1 to ST11' in Table 1, so please match the numbers in the table and figure.
We have changed S1 to S11 by ST1 to ST11, and so on in the figure´s legends.
(Same for L233, L235, L240, L286, L305, L308, L310, L311, L337, L381)
We have matched correctly the corresponding samples in the marked lines. Thank you.
7.Figure 4
As with other SEM image figures, it is desirable to have notes the structure and so on explained in the text, in the figure.
We have used Figure 3 to illustrate the most visible tissues applicable to the rest of the figures. Do you refer to the different tissues in the images? However, it is sometimes difficult to add all this information in the least preserved cross sections, so Figure 3 is the most accessible to highlight the different tissues by layers.
- Discussion section
The descriptions were very large and it were difficult to understand which data each description refers to, so we strongly recommend that you include citations to figures and tables as well as sample numbers. Also, readability of the manusctipt will be improved if the notation will be more concise, such as by avoiding duplication with the Results section.
We have organized better the discussion section following these instructions.
- Materials and Methods section
The method of statistical analysis should be added.
We have highlighted in yellow the statistical analysis in 4.2. Inductively coupled plasma mass spectrometry (ICP-MS) section.
- Location of Conclusion section
The Conclusion clause should be locataed immediately after the Discussion section, not after Materials and Methods section.
We have placed the conclusion after discussion. Thank you.
Reviewer 3 Report
Comments and Suggestions for Authors
Halophyte (salt-loving) plants represent valuable underutilized plant resources for the development of dryland agriculture. The halophytic farming system is a relatively new approach to managing salinity. It develops cropping systems for saline environments, using the capacity of halophytes and salt-tolerant non-conventional crops. It also produces good quality seeds under saline conditions via improving soil/water quality and natural agroecosystem function. In this context the article is actual. Arthrocnemum species including different populations of Mediterranean-Atlantic European and North African populations used in this study demonstrated the feasibility of this concept.
The results and conclusions of this article are well complemented by the integrative and complex methodological approach (ionic, nutritional, and biochemical analysis coupled with X-ray, TEM, and SEM studies).
Author Response
REVIEWER 3 (in black our response)
Halophyte (salt-loving) plants represent valuable underutilized plant resources for the development of dryland agriculture. The halophytic farming system is a relatively new approach to managing salinity. It develops cropping systems for saline environments, using the capacity of halophytes and salt-tolerant non-conventional crops. It also produces good quality seeds under saline conditions via improving soil/water quality and natural agroecosystem function. In this context the article is actual. Arthrocnemum species including different populations of Mediterranean-Atlantic European and North African populations used in this study demonstrated the feasibility of this concept.
The results and conclusions of this article are well complemented by the integrative and complex methodological approach (ionic, nutritional, and biochemical analysis coupled with X-ray, TEM, and SEM studies).
Thank you very much for your positive comments on our manuscript.
Reviewer 4 Report
Comments and Suggestions for Authors
Arthrocnemum Moq.: Unlocking Opportunities for Biosaline Agriculture and Improved Human Nutrition: In this article, authors studied the elemental content and biomineralization processes of two halophytic species. Overall, it looks interesting and the conclusions are clear, which highlights the potential use of Arthrocnemum species. However, there are still some problems in the article that need to be revised, such as, first, why the identification method of multiple comparisons is not the letter ABCD in tables, and the way in the article seems to be laborious. Secondly, the description of the introduction and results was not concise enough and it was suggested that they be deleted. Third, some diagrams are not clear enough, such as arrows in images of chloroplasts.
Author Response
REVIEWER 4 (in black our response)
Arthrocnemum Moq.: Unlocking Opportunities for Biosaline Agriculture and Improved Human Nutrition: In this article, authors studied the elemental content and biomineralization processes of two halophytic species. Overall, it looks interesting and the conclusions are clear, which highlights the potential use of Arthrocnemum species. However, there are still some problems in the article that need to be revised, such as, first, why the identification method of multiple comparisons is not the letter ABCD in tables, and the way in the article seems to be laborious. Secondly, the description of the introduction and results was not concise enough and it was suggested that they be deleted. Third, some diagrams are not clear enough, such as arrows in images of chloroplasts.
We have employed statistical methods to compare different geographic packages for the same element, considering only valid options a, b, and c in case of differences among all. As significant statistical differences have typically been observed only between two geographic packages at most, only options a and b have been utilized. One of the chloroplast images has been replaced with more visible arrows, and the text has been modified based on your review and the rest of the reviewers' comments in each section. Thank you for the review.
Round 2
Reviewer 2 Report
Comments and Suggestions for Authors
To authors,
I read the revised manuscript. Readability has improved considerably compared to the previous.
It also seems that you have carefully considered and corrected the comments pointed by me. In particular, the map of the survey area added as Figure 12 is very helpful in understanding this report.
On the other hand, the relationship between the survey site and the Tinto River, which the authors argue is an important geographical factor, remains unclear. Is it possible to add the Tinto River to Figure 12?
Author Response
Dear reviewer,
we have changed the map by providing Figure 12 with 3 sections to display the Tinto River's influence on some of the samples:
Figure 12. A. Map showing the distribution of Arthrocnemum macrostachyum (green circles) and Arthrocnemum meridionale (red circles) samples. B. Main area of study in the southwestern of the Iberian Peninsula (Tinto River, Huelva, Spain). C. Tinto River marshes samples. Refer to Table S1 of the Supplementary Material for sample information and analyses.
Thank you for your valuable suggestions.
Round 3
Reviewer 2 Report
Comments and Suggestions for Authors
I confirmed the revised manuscript.
I would like to recommend to the editor to publish this manuscript in this journal.